# Antibiofilm Activity and Biocompatibility of Temporin-SHa: A Promising Antimicrobial Peptide for Control of Fluconazole-Resistant *Candida albicans*

**DOI:** 10.3390/microorganisms12010099

**Published:** 2024-01-04

**Authors:** Luana Mendonça Dias, Eduardo Maffud Cilli, Karine Sousa Medeiros, Maria Carolina Oliveira de Arruda Brasil, Lina Maria Marin, Walter L. Siqueira, Ana Claudia Pavarina

**Affiliations:** 1Department of Dental Materials and Prosthodontics, School of Dentistry, São Paulo State University (UNESP), Araraquara 16015-050, Brazil; luana.dias@usask.ca (L.M.D.); karine_maraujo@hotmail.com (K.S.M.); 2College of Dentistry, University of Saskatchewan, Saskatoon, SK S7N 5E5, Canada; lina.marin@usask.ca; 3Department of Biochemistry and Organic Chemistry, Institute of Chemistry, São Paulo State University (UNESP), Araraquara 14800-060, Brazil; eduardo.cilli@unesp.br (E.M.C.); mcoa.brasil@unesp.br (M.C.O.d.A.B.)

**Keywords:** *Candida albicans*, biofilm, antimicrobial peptides, fibroblasts, keratinocytes

## Abstract

The aim of the study was to investigate the effect of antimicrobial peptides (AMPs) Hylin−a1, KR−12-a5, and Temporin-SHa in *Candida albicans* as well as the biocompatibility of keratinocytes spontaneously immortalized (NOK-si) and human gingival fibroblasts (FGH) cells. Initially, the susceptible (CaS—ATCC 90028) and fluconazole-resistant (CaR—ATCC 96901) *C. albicans* strains were grown to evaluate the effect of each AMP in planktonic culture, biofilm, and biocompatibility on oral cells. Among the AMPs evaluated, temporin−SHa showed the most promising results. After 24 h of Temporin-SHa exposure, the survival curve results showed that CaS and CaR suspensions reduced 72% and 70% of cell viability compared to the control group. The minimum inhibitory/fungicide concentrations (MIC and MFC) showed that Temporin−SHa was able to reduce ≥50% at ≥256 µg/mL for both strains. The inhibition of biofilm formation, efficacy against biofilm formation, and total biomass assays were performed until 48 h of biofilm maturation, and Temporin-SHa was able to reduce ≥50% of CaS and CaR growth. Furthermore, Temporin−SHa (512 µg/mL) was classified as non-cytotoxic and slightly cytotoxic for NOK-si and FGH, respectively. Temporin−SHa demonstrated an anti-biofilm effect against CaS and CaR and was biocompatible with NOK-si and FGH oral cells in monolayer.

## 1. Introduction

*Candida albicans* is a commensal fungus and inhabits various locations of the body, including the gastrointestinal system, genitals, and skin. Due to immunological imbalance or in immunocompromised individuals, *C. albicans* may proliferate, penetrate the superficial epithelium, and cause several fungal diseases [1,2]. The Centers for Disease Control and Prevention (United States) currently classify *C. albicans* as the third most commonly isolated blood-stream pathogen in hospitalized patients, with a mortality rate of up to 50% [3]. Treatment-wise, conventional azole and polyene antifungals are commonly used to control the infection caused by *C. albicans* [4,5]. However, *C. albicans*’ adaptative behavior enables it to form biofilms, and this mechanism of survival, especially in established biofilms, creates challenges in treatment [5].

Biofilm structures are communities of microbial populations incorporated into an extracellular matrix (ECM) of polysaccharides that provide structural support and protection for biofilm cells [2]. It has been proven in recent decades that most diseases caused by *C. albicans* are associated with the formation of biofilms on surfaces [6]. Within a biofilm, the fungi have a stable environment and can tolerate extremely high antimicrobial concentrations [7]. This resistance to antifungal drugs is a concern because failed treatment allows for the persistence or progression of the infection [5]. Antifungal resistance can present as decreased affinity/processivity of the target drug, reduced intracellular accumulation of the drug, and reduced drug effect. Specifically, the resistance mechanism is distinct depending on the mode of action of the antifungal compounds [8]. This process can be accelerated by the genetic plasticity of an organism or the existence of already-resistant/persistent strains [5]. Therefore, *C. albicans* biofilm resistance is due to multifactorial and multimolecular specificity [8]. These mechanisms may also include the expression of resistance-associated genes, the formation of mixed biofilms, and the secretion of extracellular vesicles [9].

The challenge of antimicrobial resistance to conventional medicines is evident globally, and persistent/recurrent infections are a public health concern [10,11]. To combat this issue, several alternative treatments, such as the use of antimicrobial peptides (AMPs), have been investigated to try to control the growth of microorganisms involved in these infections [12,13,14,15,16]. AMPs are short sequences of amino acids produced by all organisms that act as primary defenses against a broad spectrum of pathogens [17]. The mechanism of action of AMPs involves the presence of cationic residues that positively charge their structure; this characteristic influences their interaction with a microorganism’s cell membrane [13]. AMPs also have a high percentage of hydrophobic amino acids in their structure, which facilitates penetration into the membrane; they align themselves with the lipid core in order to weaken this structure, resulting in cell lysis [13,14].

AMPs have demonstrated antibacterial activity against Gram-negative and Gram-positive bacteria by acting on the cell membrane [12]. AMPs from amphibians have been explored and investigated, including Hylin−a1 (IFGAILPLALGALKNLIK-NH_2_). Extracted from the cutaneous secretions of a South American frog, *Hypsiboas albopunctatus*, Hylin−a1 shows broad-spectrum antimicrobial activity against Gram-negative and Gram-positive bacteria and *Candida* spp. (*C. albicans*, *Candida krusei*, *Candida parapsilosis*, and *Cryptococcus neoformans*) [18]. Temporin−SHa (FAKGIAGMAKLF-NH_2_) is another amphibian AMP that is produced by an innate immune response of the North African frog *Pelophylax saharicus.* It is extracted from the frog’s skin and has broad antimicrobial activity against Gram-positive bacteria and even against parasites such as Leishmania [12,19]. KR−12-a5 (KRIVKLILKWLR-NH_2_) is human-derived; it is a bioactive derivate of LL-37 with only 12 amino acid residues and is classified as one of the smallest AMPs with antimicrobial activity [17]. These AMPs have all shown antimicrobial action; however, the studies were restricted to planktonic cultures, and, to date, antimicrobial activity in fluconazole-resistant *C. albicans* strains has not been studied.

Considering that the mechanism of protection and survival of *C. albicans* is related to biofilm, and biofilm is commonplace in the oral cavity, the present study aimed to evaluate the antimicrobial effect of these specific AMPs on susceptible (CaS—90028) [20] and fluconazole-resistant (CaR—96901) [20] *C. albicans* strains. This included cell viability analysis (CFU/mL), survival curve, minimum inhibitory concentration (MIC) and minimum fungicide concentration (MFC), quantification of total biomass, and efficacy of AMPs in the inhibition of biofilm formation and against the biofilm formed using biofilm inhibition concentration (BIC-2). In addition, the cytotoxic effect of the AMPs on oral cells was evaluated.

## 2. Materials and Methods

### 2.1. Synthesis, Cleavage, and Purification of Peptides Hylin−a1, KR−12-a5, and Temporin-SHa

Hylin−a1, KR−12-a5, and Temporin−SHa were synthesized by solid-phase peptide synthesis using the Fmoc group protocol on Rink Amide Resin [21,22]. N,N’-diisopropylcarbodiimide (DIC), hydroxybenzotriazole (HOBt), and Fmoc-amino acids were used in a ratio of three times the number of reactive groups in the initial resin. The Fmoc group was removed using a 20% solution of 4-methylpiperidine in dimethylformamide (DMF) between 1 and 20 min. Between each step, six washes with DMF were performed. The cleavage was performed using a solution of 95% trifluoroacetic acid (TFA), 2.5% triisopropylsilane (TIS), and 2.5% ultrapure water under moderate agitation for two hours at room temperature. The peptide was then precipitated by washing three times with chilled ethyl ether. The precipitate obtained was extracted from the resin with an aqueous solution containing 0.045% TFA in water (solution A) and 0.036% TFA in acetonitrile (solution B) (1:1 V:V). The supernatant containing the peptide was lyophilized [21,22]. 

The purification of the crude peptides was performed by high-performance liquid chromatography (HPLC) in reverse phase in semi-preparative mode using a C18 AAPPTEC column (25 cm × 1 mm and 5 μm particle) for detection of 220 nm in solutions A and B with a flow rate of 5 mL/min (Figure 1). The material obtained was lyophilized and analyzed with analytical HPLC in an Ultraspehere Phenomnex C18 column (size 4.6 × 250 mm, particle size 5 μm) with detection of 220 nm using a gradient of 5 to 95% of solution B in 30 min with a flow rate of 1 mL/min. A mass spectrometer, the Thermo LCQ-Fleet with configuration ESI-IT-MS, was used to obtain the molecular weight of the synthesized material and confirm the desired peptide (Figure 2). A direct infusion of the sample solution was performed with a flow rate of 5.0 µL/min, and the electrospray source was operated in a positive mode, applying 4.5 kV to the electrospray capillary [21]. 

### 2.2. Antimicrobial Activity of AMPs on Planktonic Cultures of C. albicans

#### 2.2.1. Strains, Microorganism Growth, and Culture Conditions

Standard strains of *C. albicans* [ATCC 90028 (CaS) [20] and ATCC 96901 (CaR) [20] stored at −80 °C were thawed and seeded in petri plates with culture medium Sabouraud dextrose agar (SDA) supplemented with chloramphenicol (0.05 g) and incubated (37 °C/48 h). Then, five colonies were inoculated in 10 mL of yeast nitrogen base (YNB) medium supplemented with 100 mM of glucose and incubated (37 °C/16 h). After, 500 μL of the cultures were transferred to new falcon tubes containing 9.5 mL of YNB medium (20:1) and incubated until reaching the middle of their exponential growth phase (CaS: OD_540nm_ = 0.55 ± 0.08; CaR: OD_540nm_ = 0.91 ± 0.02) [23], corresponding to a microbial concentration of 10^7^ CFU/mL. Cultures were centrifuged and washed with phosphate-buffered saline solution (PBS, 57 mM NaH_2_PO_4_, 50 mM Na_2_HPO_4_, and 200 mM NaCl; pH 7.33) at 4000× *g* for 5 min (4 °C) and resuspended in the PBS solution. These cells were used for assays in planktonic cultures and biofilms.

#### 2.2.2. Survival Curve, Minimum Inhibitory Concentration (MIC), and Minimum Fungicide Concentration (MFC) of AMPs on Planktonic Cultures of *C. albicans*

To determine the survival curve of CaS and CaR in the presence of AMPs, the peptides were diluted in PBS, and concentrations included 2, 4, 8, 16, 32, 64, 128, 512, and 1024 μg/mL. The AMPs were added to a 96-well plate containing the inoculum (adjusted 10^7^ CFU/mL^−1^) and incubated aerobically at 37 °C. After incubation times of 0, 5 min, 10 min, 20 min, 30 min, 1 h, 2 h, 3 h, 4 h, 6 h, and 24 h, aliquots of 0.01 mL were plated in SDA to determine CFU/mL [24]. To determine the MIC of AMPs required to inhibit the visual microbial growth of *C. albicans* [21], the test was performed according to the Clinical and Laboratory Standards Institute (CLSI). The contamination control was 100 μL of Roswell Park Memorial Institute (RPMI) medium 1640 (Sigma-Aldrich, St. Louis, MO, USA) and 100 μL of PBS. The growth control was 100 μL of cell suspension without an antimicrobial agent. Hylin−a1, KR−12-a5, and Temporin−SHa (ranging from 2 to 1024 μg/mL) were inoculated in 24-well plates with 100 μL of 0.5 × 10^3^ to 2.5 × 10^3^ CFU/mL of CaS and CaR in RPMI 1640, buffered with 3-N-Morpholino propanosulfonic acid (MOPS) buffer 0.65 M (pH 7.0). After 24 h of incubation, the plates were visually observed for the presence or absence of fungal growth to determine the lowest concentration of AMP to inhibit growth [25,26]. To determine the MFC, aliquots of 0.01 mL were plated in SDA to determine CFU/mL [27]; the concentration that reduced 50% of viable colonies compared to the AMP-free control was considered the MFC [26]. Figure 3 contains the flowchart for the steps previously described. All experiments were performed in quadruplicate on three different occasions (*n* = 12).

### 2.3. Cytotoxicity Evaluation of Hylin−a1 and KR−12-a5 on Oral Cell Cultures in Normal Oral Keratinocytes (NOK-si) and Human Gingival Fibroblasts (FGH)

#### 2.3.1. Cell Line and Cell Culture

NOK-si (kindly provided by Professor Carlos Rossa Jr., from the Cellular and Molecular Biology Laboratory, Department of Periodontics, School of Dentistry, São Paulo State University—UNESP) and FGH (Rio de Janeiro Cell Bank; code: 0089) cells were thawed and cultured in Eagle medium modified by Dulbecco high glucose (DMEM) (4.5 g/L), supplemented with 2.0 mmol·L^−1^ of glutamine (Lonza, Basel, Switzerlnad, 10% of bovine serum), 1% antibiotic/antimycotic (penicillin G—10,000 μg·mL^−1^, streptomycin—10,000 μg·mL^−1^, amphotericin B—25 μg·mL^−1^) (Sigma-Aldrich, St. Louis, MO, USA) incubated with 5% CO_2_ in the atmosphere and cultured until reaching 80% confluence. The cells were washed (PBS buffer; pH 7.2), detached from the plate with a 0.05% trypsin ethylenediaminetetraacetic acid (EDTA) solution (Sigma-Aldrich, St. Louis, MO, USA), and then the cells were counted. In the experiments, 2.5 × 10^5^ cells/well were used for NOK-si and 1.2 × 10^5^ cells/well were used for FGH. For all experiments, cells between the 3rd and 8th passages were used [28].

#### 2.3.2. Cell Viability Analysis by alamarBlue™

To elucidate the biocompatibility of AMPs with NOK-si and FGH oral cells, a volume of 200 μL of cells resuspended in DMEM was plated in 96-well plates and incubated (37 °C/5% CO_2_/16 h). Cells adhered to the bottom of the plate were washed and exposed to AMPs at concentrations of 32, 64, 128, 256, and 512 μg/mL. These concentrations were previously determined in the MIC/MFC evaluation. After, the solution consisting of 10% alamarBlue™ and 90% of DMEM medium supplemented with fetal bovine serum (FBS) was added, incubated for up to 48 h, and read using Fluoroskan Ascent™ (FL, Thermo Scientific, Waltham, MA, USA; excitation—544 nm; emission—590 nm). NOK-si and FGH cells in the absence of treatment were used as live controls. Triton x-100 (0.9%—BRAND) solution was used as a death control culture. The results obtained were normalized and classified with respect to cytotoxicity according to ISO 10993-5:2009 guidelines [29] (Table 1).

### 2.4. Antimicrobial Activity of AMPs against C. albicans Biofilm 

#### 2.4.1. Quantification of Total Biofilm Biomass of *C. albicans* (CaS and CaR) with Crystal Violet Dye

For quantification of the total biofilm biomass of CaS and CaR biofilm, 100 μL of the inoculum at the final concentration of 10^7^ CFU/mL was transferred to 96-well plates containing 100 μL of YNB supplemented with glucose at 100 mM. These plates were incubated (37 °C/75 rpm) for 1.5 h, corresponding to the adhesion phase, to allow the cells to adhere to the well plate bottom, followed by washing in PBS solution to remove non-adhered or loosely adhered cells [30]. Then, YNB medium with AMPs at concentrations of 32, 64, 128, 256, and 512 μg/mL was inserted in the wells. The plates were incubated at 37 °C, and when biofilm formation reached 24 h, 100 μL of the medium was removed via aspiration and 100 μL of the same fresh medium was added. After 48 h of biofilm maturation in the presence of the AMPs, the biofilms were washed (3× PBS) and fixed with methanol (200 μL/15 min). Then, the biofilms were dried for 20 min, and 200 μL of 1% crystal violet dye was inserted. After 5 min, 200 μL of 33% acetic acid was added to remove the dye. Finally, aliquots were transferred to a 96-well plate for quantification of total biomass using a spectrophotometer (570 nm). Biofilms formed in the absence of AMPs were used as a control. The experiments were performed in quadruplicate on three different occasions (*n* = 12/group).

#### 2.4.2. Efficacy of AMPs in the Inhibition of Biofilm Formation and against the Biofilm Formed

Two 24-well plates were used, one to determine the inhibition of biofilm formation and the other to determine the efficacy of AMPs against the formed biofilm [6]. To assess inhibition of biofilm formation, a standardized suspension of CaS and CaR (500 µL) and YNB culture medium (500 µL) were inoculated into each well. The plates were incubated for 1.5 h (37 °C/75–77 rpm), corresponding to the adhesion phase. After, the plates were washed (3× PBS) and AMPs diluted in YNB (2 mL) were added to the plate for evaluation of the inhibition of biofilm formation. Six concentrations of AMPs were evaluated to determine which resulted in a significant reduction in the viability of planktonic cultures: 32, 64, 128, 256, and 512 μg/mL. After 48 h of incubation, the biofilms were detached by scraping for 45 s, followed by serial dilution (10^−1^ to 10^−4^) and plating in Petri dishes containing SDA. 

To determine the efficacy of AMPs against the biofilm already formed, the culture medium was changed (2 mL/each well) after the adhesion phase (1.5 h), and the plate was kept in an incubator for 24 h until the biofilm matured. After this period, the biofilm was washed (3× PBS) and YNB (2 mL) containing AMPs (1024, 512, 256, 128, 64, and 32 μg/mL) were added into wells. The plates were incubated for an additional 24 h. After 48 h, the biofilms were detached by scraping for 45 sec, followed by serial dilution (10^−1^ to 10^−4^) and plating in SDA. For both assays, the efficacy of treatment was considered to be the lowest AMP concentration required to reduce cell viability by 50% compared to the control group [6]. 

### 2.5. Data Analysis

The in vitro experimental analyses used CFU/mL or arbitrary units (UA—optical density through absorbance, fluorescence, and viability of oral cells) as the continuous (dependent) quantitative response variable. The nominal qualitative independent variables are concentrations of AMPs and incubation times of AMPs. When the Shapiro–Wilk test showed no normal distribution of data, Levene’s test was used, and for heteroscedastic variance, the ANOVA-one-way test with Welch’s correction was applied, followed by the Games-Howell post-test for multiple comparisons. When the Shapiro–Wilk test showed a normal distribution, the one-way ANOVA test was applied, followed by Tukey’s post-test for multiple comparisons. SPSS software (version 2.0) was used, with a significance level of 5% [31].

## 3. Results

### 3.1. Survival Curve, Minimum Inhibitory Concentration (MIC), and Minimum Fungicide Concentration (MFC) of AMPs on Planktonic Cultures of C. albicans

Figure 4a shows the log_10_ survival of CaS suspensions after longitudinal exposure (up to 24 h) to the Hylin−a1 peptide. Significant viability reduction of ≥50% compared to the control group (*p* ≤ 0.05) was observed after 6 and 24 h at concentrations of 128 µg/mL (6 h: 4 log_10_/56%; 24 h: 4.1 log_10_/58%), 256 μg/mL (6 h: 3.6 log_10_/51%; 24 h: 4.2 log_10_/59%), 512 μg/mL (6 h: 4.7 log_10_/67%; 24 h: 4.9 log_10_/69%), and 1024 μg/mL (6 h: 4.8 log_10_/68%; 24 h: 5.3 log_10_/75%). Regarding the viability of CaR suspensions (Figure 4b) after treatment with Hylin−a1, the results showed significant reduction compared to the control (*p* ≤ 0.05) after 6 and 24 h at concentrations of 256 μg/mL (6 h: 4 log_10_/56%; 24 h: 4.1 log_10_/58%), 512 μg/mL (6 h: 4.7 log_10_/67%; 24 h: 4.9 log_10_/70%), and 1024 μg/mL (6 h: 4.8 log_10_/68%; 24 h: 5 log_10_/73%). The results regarding the survival curve of CaS and CaR after exposure to Fluconazole are in Appendix A.

After longitudinal exposure (up to 24 h) of CaS (Figure 4c) and CaR (Figure 4d) to KR−12-a5, the results demonstrated that no concentrations reduced viability ≥50% for both strains. Among the concentrations, 1024 μg/mL at 24 h showed the highest viability reduction in the analysis (CaS: 2.8 log_10_/39%; CaR: 2.5 log_10_/35%), compared to the control group (*p* ≤ 0.05).

The significant viability reduction (≥50%) was observed after longitudinal exposure of CaS planktonic culture to Temporin−Sha (Figure 4e) compared to the control (*p* ≤ 0.05) after 6 and 24 h at concentrations of 128 μg/mL (6 h: 4 log_10_/56%; 24 h: 4.1 log_10_/58%), 256 μg/mL (6 h: 3.6 log_10_/51%; 24 h: 4.2 log_10_/59%), 512 μg/mL (6 h: 4.7 log_10_/67%; 24 h: 5 log_10_/70%), and 1024 μg/mL (6 h: 4.7 log_10_/67%; 24 h: 5.1 log_10_/73%). Regarding the results of CaR after the longitudinal exposure to Temporin−Sha (Figure 4f), a significant reduction (≥50%) was observed compared to the control group (*p* ≤ 0.05) after 6 and 24 h at concentrations of 256 μg/mL (6 h: 4.7 log_10_/57%; 24 h: 4.9 log_10_/58%), 512 μg/mL (6 h: 4.7 log_10_/67%; 24 h: 4.7 log_10_/69%), and 1024 μg/mL (6 h: 4.4 log_10_/67%; 24 h: 5 log_10_/70%).

The results of susceptibility tests obtained after treatment of CaS and CaR planktonic cultures with AMPs showed that, for concentrations from 0 to 128 µg/mL, visible turbidity (MIC: CLSI parameter) [22] was observed for Hylin−a1, KR−12-a5, and Temporin−Sha. Thus, the MIC assay was performed by plating the first/minimum concentration that inhibited growth (256 µg/mL) and the four adjacent concentrations (64, 128, 512, and 1024 µg/mL) to determine the MFC by CFU/mL.

For treatment with Hylin−a1, the concentrations capable of a reduction of ≥50% MFC for CaS (Figure 5a) and CaR (Figure 5b) viable colonies were ≥128 µg/mL and ≥256 µg/mL, respectively, compared to the experimental control (*p* ≤ 0.05). Additionally, the concentration of 1024 µg/mL resulted in the highest reduction of viable colonies for CaS and CaR with 5.3 log_10_ (75%) and 5.1 log_10_ (72%), respectively, when compared to the control (*p* ≤ 0.05). The results regarding the MIC and MFC of CaS and CaR after exposure to Fluconazole are in Appendix A.

For treatment with KR−12-a5, no concentrations were capable of a reduction of ≥50% for CaS (Figure 5c) and CaR (Figure 5d) viable colonies. The highest viable colony reductions of CaS and CaR were found after treatment with KR−12-a5 at a concentration of 1024 µg/mL, with 2.7 log_10_ and 2.6 log_10_ decreases, respectively. Although not capable of ≥50% reduction, this concentration was statistically different from the control group (*p* ≤ 0.05). 

For treatment with Temporin−SHa, the concentration capable of a reduction of ≥50% MFC of CaS (Figure 5e) and CaR (Figure 5f) viable colonies was ≥256 µg/mL. The highest CaS and CaR viable colony reductions were found at 1024 µg/mL, with 5.1 log_10_ (72%) and 5 log_10_ (70%) reductions for CaS and CaR, respectively, with this being statistically different from the control group (*p* ≤ 0.05).

### 3.2. Quantification of Total Biofilm Biomass of C. albicans (CaS and CaR) 

In respect to quantifying the total biomass of the CaS and CaR biofilms after exposure to AMPs, the ability of six concentrations to statistically reduce the planktonic cultures of CaS and CaR was evaluated and compared to control (*p* ≤ 0.05): 32, 64, 128, 256, 512, and 1024 µg/mL. For Hylin−a1 and KR−12-a5, in both CaS and CaR, the results showed no statistical difference among the concentrations evaluated (32 to 1024 µg/mL) and experimental control (*p* ≤ 0.05) (Figure 6a–d). 

In respect to Temporin−SHa, total biomass reduction for CaS was 54% for 1024 µg/mL concentration and 50% for 512 µg/mL concentration (Figure 6e). It was observed that 512 and 1024 µg/mL concentrations had the lowest means in the analysis and were statistically different from the control group (*p* ≤ 0.05). In the analysis of CaR total biomass after treatment with Temporin−SHa, it was observed that concentrations from 128 to 1024 µg/mL were similar to each other and different statistically from control (*p* ≤ 0.05), with the lowest means of the analysis. In addition, the concentration of 128 µg/mL promoted a 30% reduction in CaR total biomass compared to the control group (*p* ≤ 0.05) (Figure 6f).

### 3.3. Efficacy of AMPs in the Inhibition of Biofilm Formation and against the Biofilm Formed

For efficacy in inhibiting biofilm formation, the Hylin−a1 concentration of 1024 µg/mL resulted in a reduction of 2 log_10_ (28%) and 1.9 log_10_ (27%) for CaS (Figure 7a) and CaR (Figure 7b), respectively, having the lowest means in the analysis. Additionally, there was no statistical difference among the concentrations evaluated (32 to 1024 µg/mL). Regarding the efficacy of Hlyin-a1 to control biofilm that has already formed, a concentration of 1024 µg/mL resulted in a reduction of 1 log_10_ and 0.8 log_10_ for CaS (Figure 8a) and CaR (Figure 8b), respectively, compared to the control group (*p* ≤ 0.05). Therefore, there was no concentration of Hylin−a1 able to reduce BIC-2 by ≥50% with respect to CaS and CaR biofilm viability.

For the efficacy of KR−12-a5 to inhibit biofilm formation, a concentration of 1024 µg/mL resulted in a reduction of 1.9 log_10_ and 1.7 log_10_ for CaS (Figure 7c) and CaR (Figure 7d), respectively, with no statistical difference between the concentrations evaluated. Regarding the efficacy of KR−12-a5 to control biofilm that has already formed, there was a decrease of 0.8 log_10_ and 0.7 log_10_ for CaS (Figure 8c) and CaR (Figure 8d), respectively, compared to the control group (*p* ≥ 0.05). No statistical difference was also observed among the concentrations evaluated. 

For the efficacy of Temporin−SHa to inhibit biofilm formation, a concentration of 1024 µg/mL resulted in a reduction of 4.6 log_10_ and 4 log_10_ for CaS (Figure 7e) and CaR (Figure 7f), respectively, which is statistically significant compared to the control group. Regarding the efficacy of Temporin−SHa to control biofilm that has already formed at a concentration of 1024 µg/mL, there was a 3.8 log_10_ and a 3.5 log_10_ for CaS (Figure 8e) and CaR (Figure 8f), respectively. This is statistically significant compared to the control group (*p* ≥ 0.05).

### 3.4. Cytotoxicity Evaluation of Hylin−a1 and KR−12-a5 in Oral Cell Cultures on Normal Oral Keratinocytes (NOK-si) and Human Gingival Fibroblasts (FGH)

The results of the cytotoxicity analysis of the AMPs on NOK-si and FGH monolayer cells by alamarBlue showed that Hylin−a1 (Figure 9a,b) and KR−12-a5 (Figure 9c,d) reduced cell viability by ≥75%, being classified as severely cytotoxic according to ISO guidelines. There was no statistical difference between all concentrations evaluated (32, 64, 128, 256, 512, and 1024 µg/mL). 

The cytotoxicity analysis of Temporin−SHa on NOK−si cells showed that all concentrations evaluated were statistically similar to each other and to the live control (CT), resulting in Temporin−SHa being considered non-cytotoxic according to ISO standards (Figure 9e). Additionally, the concentration of 512 µg/mL resulted in a viability reduction of only 5% compared to the control group. The cytotoxicity analysis for FGH cells (Figure 9f) demonstrated that concentrations of 128 and 512 µg/mL reduced viability by 47 and 50%, respectively, compared to the control group. This results in a classification of slightly cytotoxic according to ISO guidelines.

Additionally, the Temporin−SHa data obtained from the alamarBlue assay for FGH (Figure 10a) and NOK−si (Figure 10b) cells were used to calculate the cytotoxic concentration able to reduce viability cells by 50% (CC 50). The CC50 of Temporin−SHa for FGH and NOK−si cells were concentrations of 492 µg/mL and 3805 µg/mL, respectively.

The selectivity index (SI), selectivity of cells relative to themselves, was also calculated and it was observed that Temporin−SHa was 87% more selective for FGH cell when compared to NOK−si.

## 4. Discussion

Living organisms produce natural antibiotic-like molecules that inhibit the growth of microorganisms, called AMPs [32]. These peptides hold great promise for developing novel antimicrobials with a lower susceptibility to microbial resistance [33]. In the present study, the antifungal effects of Hylin−a1, KR−12-a5, and Temporin−SHa on CaS and CaR were investigated, as well as the cytotoxicity of NOK−si and FGH oral cells.

The survival curve results demonstrated that planktonic cultures exposed to Hylin−a1 and Temporin−SHa at concentrations ≥ 128 µg/mL reduced viability by ≥50% after 6 h of exposure. On the other hand, in a previous study of *C. albicans* suspensions treated with [K3]Temporin−SHa at 22 µM [34], no *C. albicans* growth was detected after 3 h of treatment. In the present study, after 1 h of Temporin−SHa treatment, a ≅ 15% decrease in viability was observed in CaS and CaR suspensions. Although these peptides belong to the same family, direct comparison is difficult since [K3]Temporin−SHa is an analog of Temporin−SHa that contains a lysine residue in position 3 instead of a serine, explaining the divergent results [34]. Another study demonstrated that Temporin−SHa at 15.6 µM was active against the clinical isolates of *Helicobacter pylori*, resulting in 90–100% death in less than 1 h of incubation under aerobic conditions [35]. These findings differ from the present study and can be explained by differences in the characteristics of the microorganisms evaluated. Fungal cells are 10× larger than bacterial cells, which makes controlling the biofilm more difficult [2]. Furthermore, *C. albicans* presents a complex cellular structure that can form multicellular structures, such as mycelium, which contribute to cell survival. On the contrary, bacteria are unicellular organisms with a simple cellular architecture containing cell membrane, cytoplasm, and cell wall [36]; this structure makes them easier to eliminate. Furthermore, fungal cell walls are made of chitin, which is more difficult to penetrate than the peptidoglycan walls present in Gram-positive bacteria [37]. Regarding the effect of Hylin−a1 against *C. albicans*, the results after 24 h showed that the highest concentration of 1024 μg/mL reduced cellular viability by 73%. A previous study showed that the hemolytic activity of *C. albicans* suspensions was reduced by 83% after 24 h of treatment with Hylin−a1 [18].

The MIC test determines the lowest antimicrobial concentration able to inhibit the visible growth of the target microorganism [25]. In this study, the broth microdilution test was used to analyze the MIC, followed by an MFC assay to confirm the *C. albicans* growth inhibition. After plating, the results showed that Temporin−SHa and Hylin−a1 at concentrations ≥256 µg/mL were able to reduce planktonic culture viability by ≥50%. In comparison, two previous studies investigated the fluconazole MIC against the same *Candida* strain (*C. albicans* ATCC 96901) used in the present study, and the authors found that suspensions of fluconazole-resistant *C. albicans* had a concentration of 256 μg/mL as the MIC value [38,39]. In another study, no growth of *C. albicans* was observed after 24 h of incubation with [K3]Temporin−SHa at a concentration of 22 µM [34]. The distinction between the findings can be explained by the AMP synthesis method, molecular conformation, and MIC methodology guidelines. In the present study, Temporin−SHa was carefully purified by HPLC in reverse phase in the semi-preparative mode method [21], and the broth microdilution method (CLSI) was performed, which is considered the most accurate assay for *Candida* spp. antifungal susceptibility due to its standardized procedure [26]. On the other hand, Brunet et al., 2022, used the European Committee on Antimicrobial Susceptibility Testing (EUCAST) guidelines to determine MIC values. Regarding KR−12-a5, the results of this study showed there were no concentrations able to reduce CaS and CaR suspension viability by ≥50%. A previous study performed a MIC assay with KR−12-a5 using a different strain of *C. albicans* (ATCC 26790), and a 15.62 μg/mL^−1^ concentration was able to inhibit 100% of *Candida* growth [40]. The discrepancy between these results could be related to the difference in the *Candida* strains used. 

Although *C. albicans* suspensions are found in the oral cavity, it is well established that the predominant culture in the oral microbiome is biofilm [41]. Microorganisms in biofilms are between 10 and 1000 times more resistant to antimicrobials than when planktonic [33,35,42]. Therefore, investigating whether antimicrobial agents are able to penetrate biofilm and break down this three-dimensional structure has been a challenge in the control of pathogenic microorganisms. Among the peptides tested, only Temporin−SHa inhibited biofilm formation by ≥50% for CaS and CaR exposed to AMP at the beginning of the adhesion phase. In addition, Temporin−SHa reduced the biofilm already formed by ≥50% for CaS and CaR exposed to AMP after 24 h of biofilm maturation. Temporin−SHa also showed a significant reduction in the total biofilm biomass of CaS and CaR. Considering that the penetration mechanism of AMPs is directly related to the composition of the microorganism cell membrane, this study suggests that Temporin−SHa was able to penetrate this structure and induce the formation of pores or/and channels in the hydrophobic core of the eukaryotic bilayer of this fungus, weakening these components and promoting cell lysis, as similarly proven in a previous study [37]. The action of other AMP classes to control *C. albicans* biofilm has been observed in the literature. AMP-17 peptide showed strong efficacy in *C. albicans* (SC 5314), inhibiting 86% of mature biofilm [43]. Additionally, the gH625-M membranotropic peptide showed efficacy, inhibiting 52% of persistently derived *C. albicans* (ATCC 90028) by targeting the structure of the cell membrane [44]. Given the cationic action of AMPs noted in the literature, it can be suggested that Temporin−SHa was able to disrupt the morphology and permeability of microbial cells to damage the structure and reach the target cell. This innovative study is the first to analyze AMPs, such as Temporin−SHa, in a tri-dimensional model studying biofilm culture. 

Despite the promising results of Hylin−a1 treatment in CaS and CaR suspensions, no reduction in biofilm culture growth was observed. These results can be explained by the low absorption of this antimicrobial agent by the *C. albicans* biofilm matrix. Although peptides from the Hylin−a1 family have been shown to interact with lipid bilayers of bacteria biofilms [45], the Hylin−a1 net charge may not have been able to penetrate the *C. albicans* biofilm multilayer and reach the target cell [26,46].

To ensure the safe use of AMPs as antifungal agents, it is also necessary to evaluate the biocompatibility of the antimicrobial agents with host cells. Among the peptides evaluated on NOK-si and FGH oral cells using the alamarBlue assay, Temporin−SHa showed the most promising results. All concentrations of Temporin−SHa evaluated on NOK-si were biocompatible, with a CC50 of 3805.5 µg/mL. For FGH, viability reduction was ≅50% and classified as slightly cytotoxic (according to ISO guidelines), with a CC50 of 492.4 µg/mL. The sensitivity of FGH could be explained by its lineage in the oral tissue architecture as a first contact cell, compared to NOK-si, which is found in the most underlying layer [28,35]. On the other hand, in a previous study, authors performed the cell viability assay with a resazurin-based kit (alamarBlue) and observed that Temporin−SHa at 1558 ± 324 µM was cytotoxic on stomach cells (N87) [35]. These results can be explained by a difference in pH, since gastric cells experience a permanently acidic environment of ≅1.5–2 pH [35], while oral cavity cells experience ≅ 6–7 pH in normal conditions [37]. 

## 5. Conclusions

The AMPs Hylin−a1 and Temporin−SHa promoted over 50% of antifungal activity inhibition against CaS and CaR suspensions; however, among the peptides evaluated in this study, only Temporin−SHa has an anti-biofilm effect in both CaS and CaR strains and is biocompatible with NOK-si and FGH oral cells in monolayer. The results described here provide evidence of Temporin−SHa as a potential and safe AMP to control fluconazole-resistant *C. albicans* biofilm. Given that the present study was limited to monolayers, an investigation of the 3D and in vivo models should be explored.

## Figures and Tables

**Figure 1 microorganisms-12-00099-f001:**
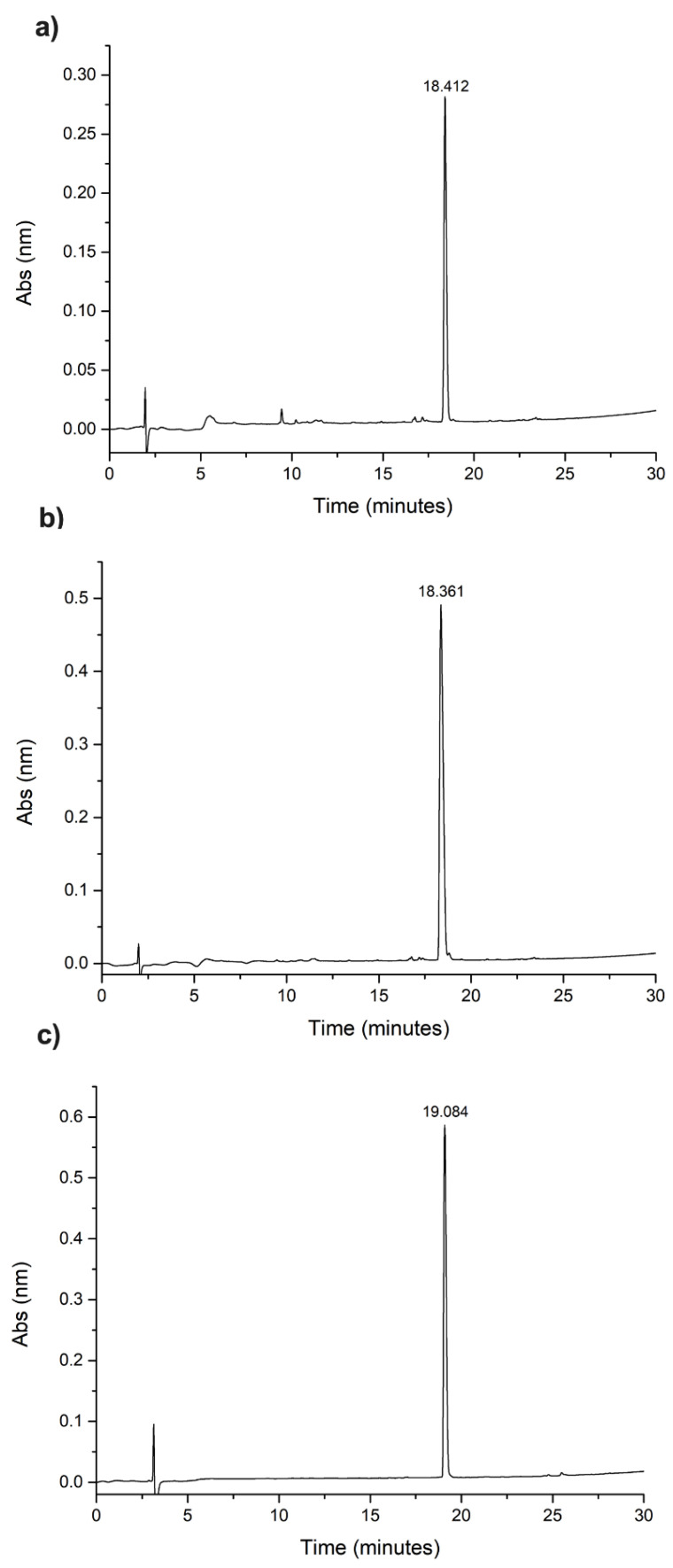
(**a**) RP-HPLC chromatogram of pure Hylin−a1; (**b**) RP-HPLC chromatogram of the pure KR−12-a5; (**c**) RP-HPLC chromatogram of the pure temporin-SHa. C18 column (25 cm × 10 mm), detection at 220 nm, using a gradient method from 5 to 95% of solvent B in 30 min with a flow rate of 1 mL min^−1^.

**Figure 2 microorganisms-12-00099-f002:**
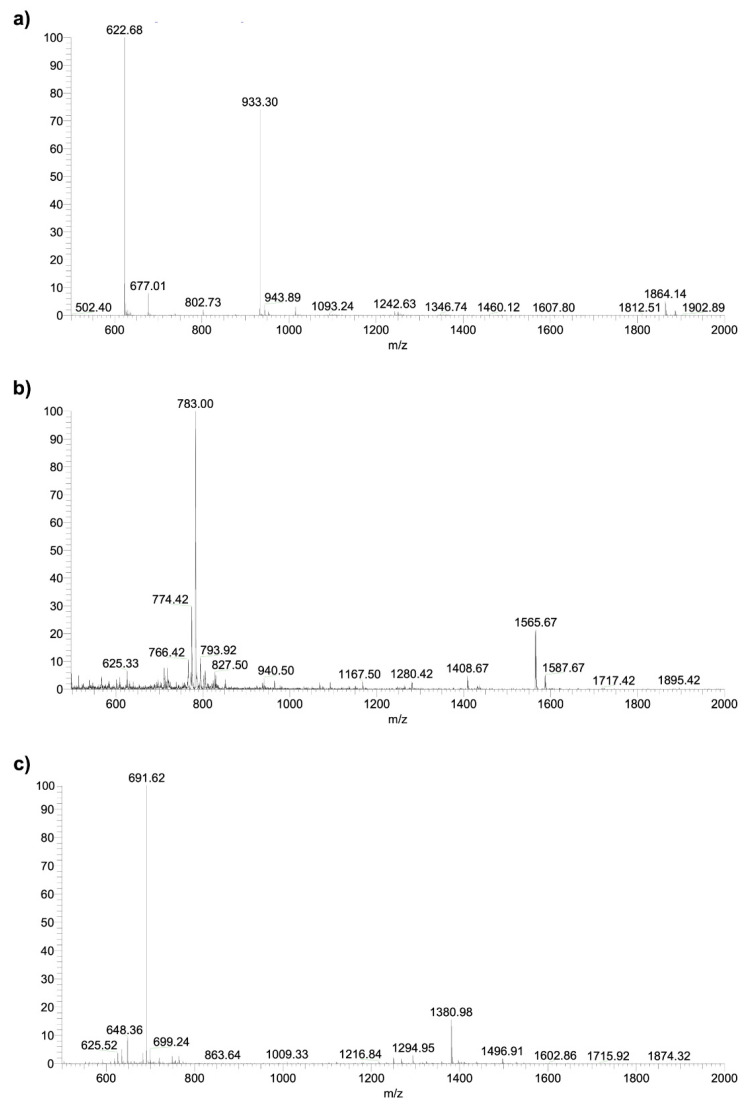
(**a**) Mass spectrum of pure Hylin−a1. Theorical MW = 1864.4 g mol^−1^; obtained MW/Z = 1864.9 (Z = 1); 933.3 (Z = 2); and 622.6 (Z = 3); (**b**) Mass spectrum of pure KR−12-a5. Theorical MW = 1565.1 g mol^−1^; obtained MW/Z = 1564.7 (Z = 1); and 783.0 (Z = 2); (**c**) Mass spectrum of pure Temporin−SHa. Theorical MW = 1380.8 g mol^−1^; obtained MW/Z = 1380.8 (Z = 1); and 691.2 (Z = 2).

**Figure 3 microorganisms-12-00099-f003:**
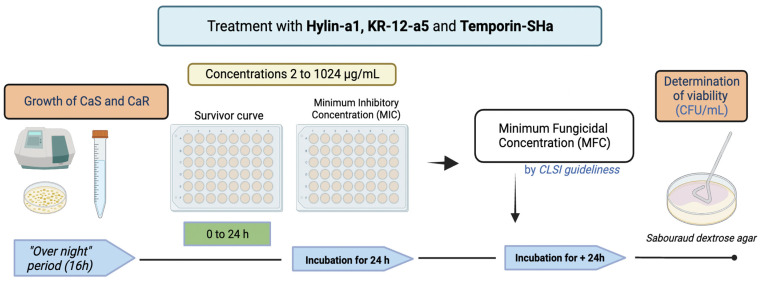
Flowchart of treatments with Hylin−a1, KR−12-a5, and Temporin−SHa: Survivor curve, minimum inhibitory concentration (MIC), and minimum fungicidal concentration (MFC).

**Figure 4 microorganisms-12-00099-f004:**
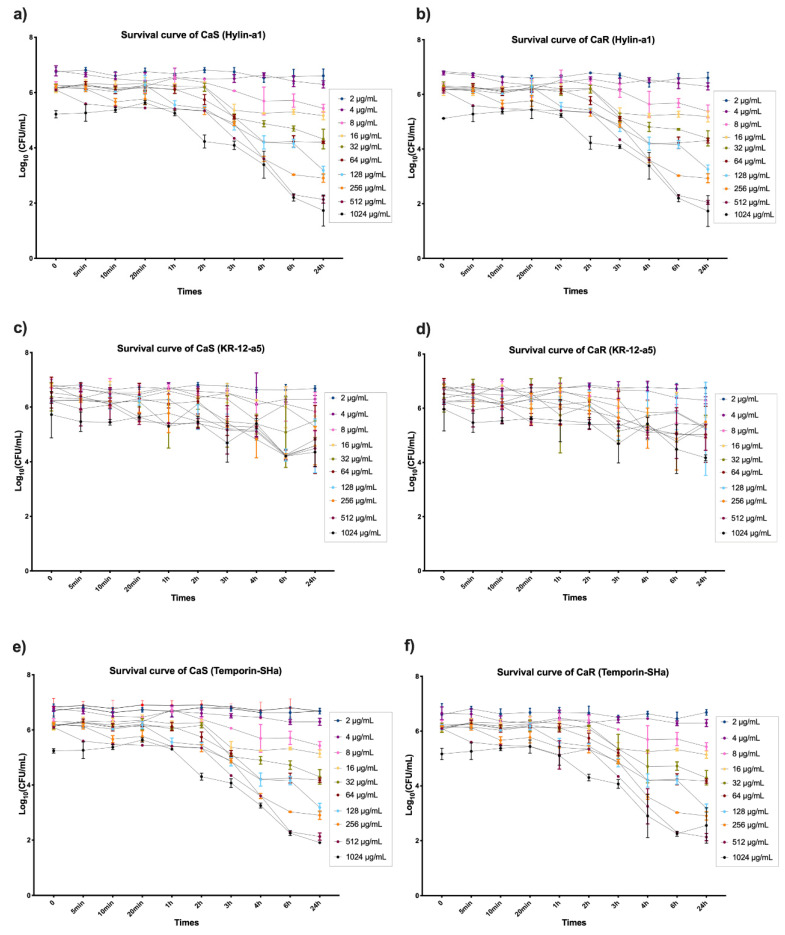
Mean and 95% confidence interval of survival in log_10_ of CaS and CaR suspensions after longitudinal exposure to predetermined concentrations (2, 4, 8, 16, 32, 64, 128, 512, and 1024 μg/mL) of Hylin−a1 (**a**,**b**), KR−12-a5 (**c**,**d**) and Temporin−Sha (**e**,**f**) at predetermined times (0; 5 min; 10 min; 20 min; 30 min; 1 h; 2 h; 3 h; 4 h; 6 h; and 24 h). Points: data averages. Error bars: minimum and maximum values. The non-intersection of the error bars denotes a statistical difference according to the 95% confidence interval (*n* = 12/group; *p* < 0.05).

**Figure 5 microorganisms-12-00099-f005:**
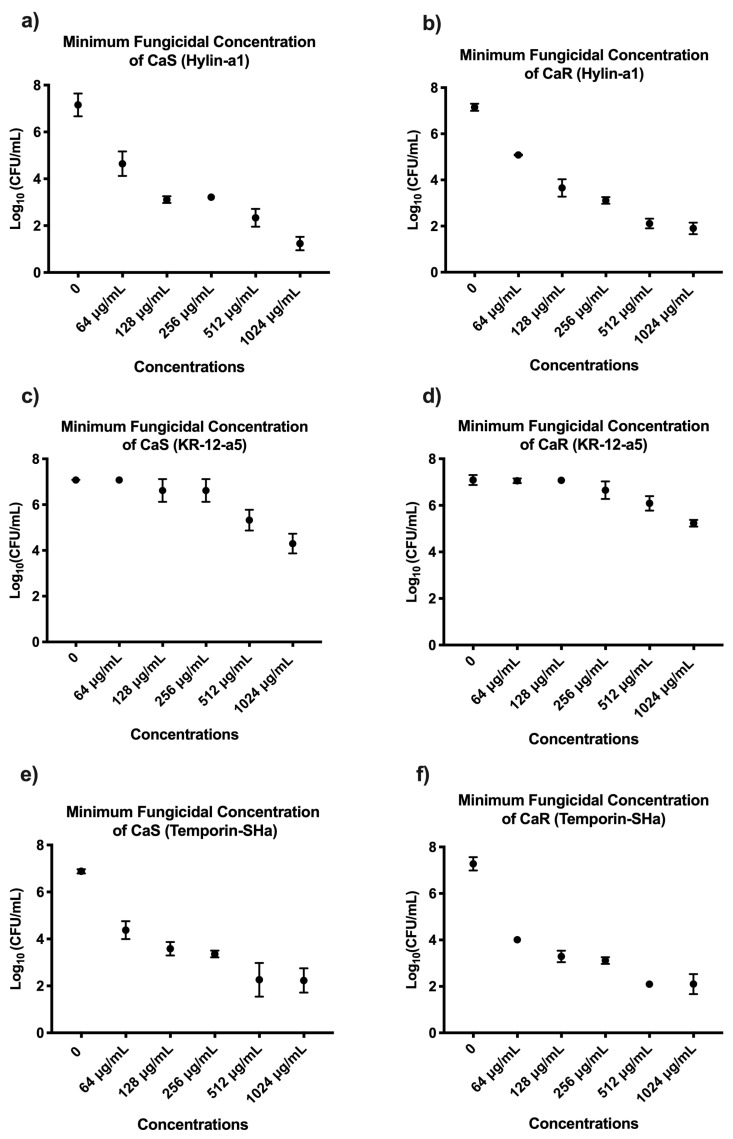
Mean and 95% confidence interval of the MFC (log_10_) of CaS and CaR suspensions after exposure to Hylin−a1 (**a**,**b**), KR−12-a5 (**c**,**d**), and Temporin−SHa (**e**,**f**) at 64, 128, 256, 512, and 1024 µg/mL. Points: data means. Error bars: minimum and maximum values. The non-intersection of the error bars denotes a difference according to the 95% confidence interval (*n* = 12/group; *p* < 0.05).

**Figure 6 microorganisms-12-00099-f006:**
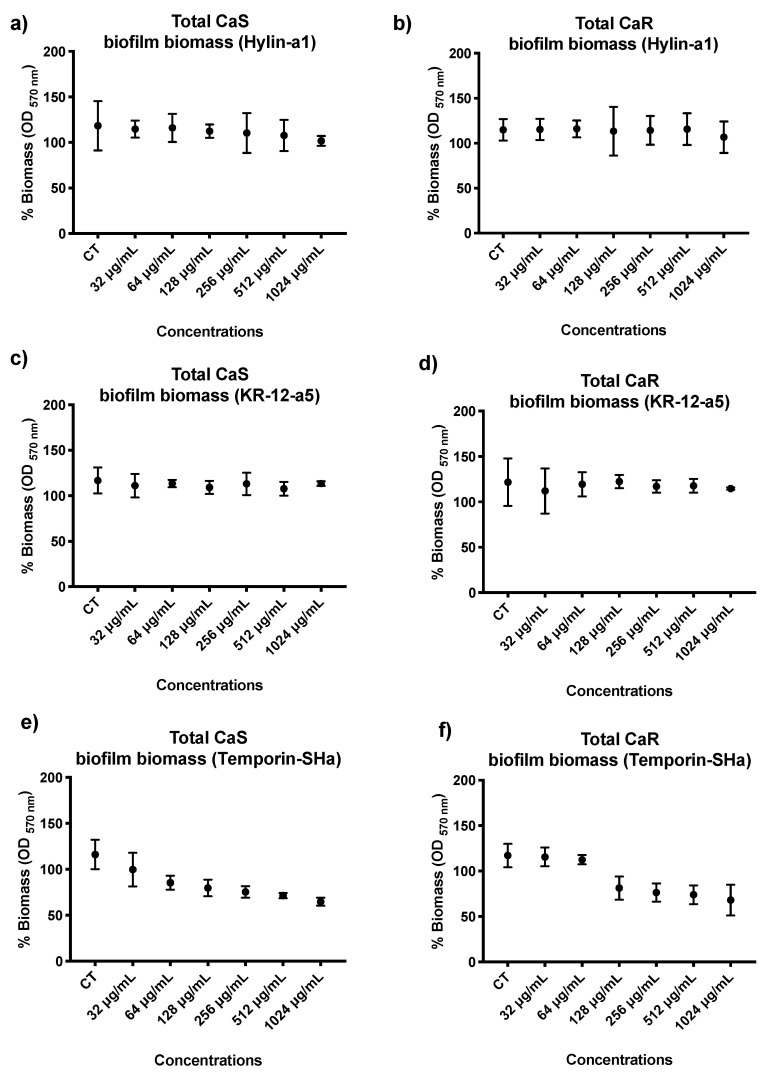
Mean and 95% confidence interval of CaS and CaR biomass (percentage: %) measured through 570 nm of optical density (OD) after exposure to Hylin−a1 (**a**,**b**), KR−12-a5 (**c**,**d**), and Temporin−SHa (**e**,**f**) at 32, 64, 128, 256, 512, and 1024 µg/mL. CT: experimental control. Points: data averages. Error bars: minimum and maximum values. The non-intersection of the error bars denotes a statistical difference according to the 95% confidence interval (*p* < 0.05; *n =* 12/group).

**Figure 7 microorganisms-12-00099-f007:**
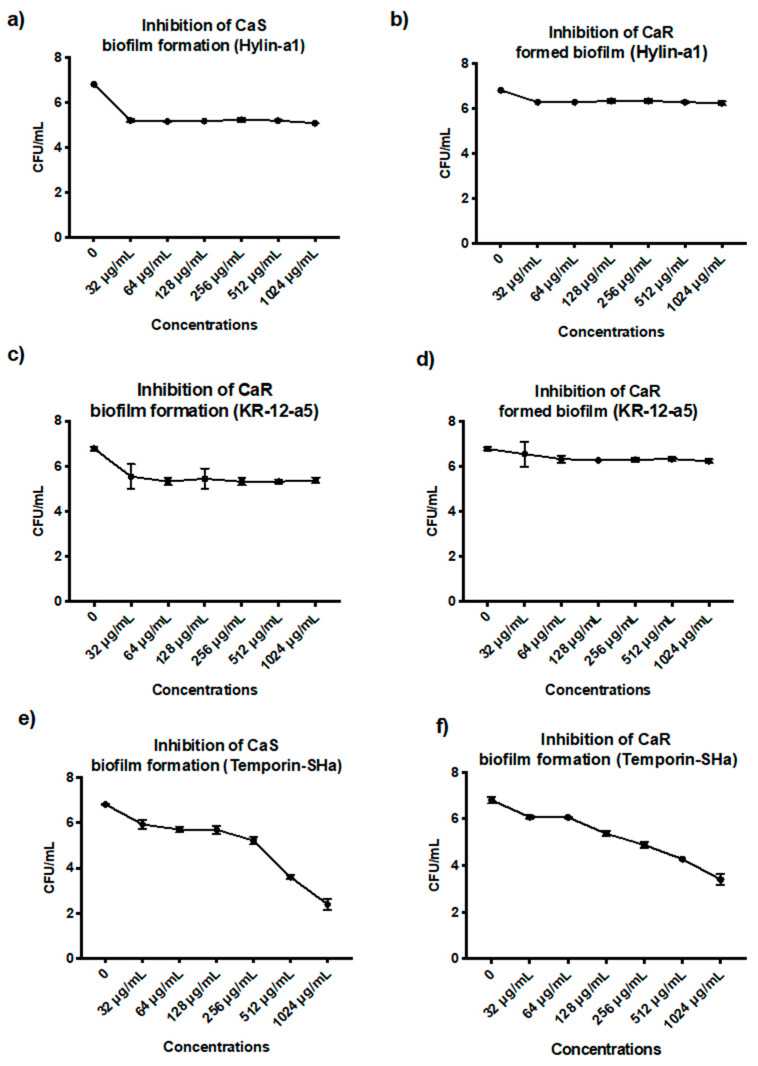
Mean and 95% confidence interval of the inhibition of CaS and CaR biofilm formation after exposure to Hylin−a1 (**a**,**b**), KR−12-a5 (**c**,**d**), and Temporin−SHa (**e**,**f**). AMPs at 32, 64, 128, 256, 512, and 1024 µg/mL and experimental control (0). Points: data averages. Error bars: minimum and maximum values. The non-intersection of the error bars denotes a statistical difference according to the 95% confidence interval (*p* < 0.05) (*n =* 12/group).

**Figure 8 microorganisms-12-00099-f008:**
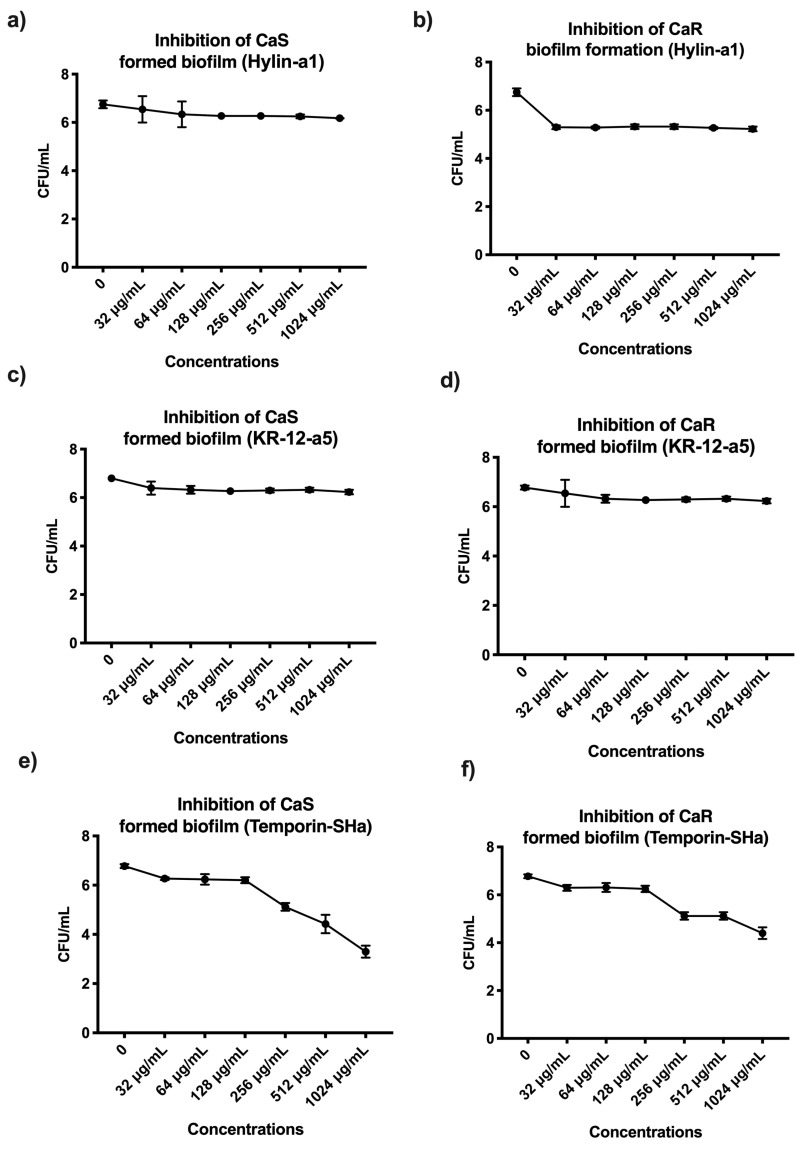
Mean and 95% confidence interval of Inhibition of CaS and CaR biofilm formed after exposure to Hylin−a1 (**a**,**b**), KR−12-a5 (**c**,**d**) and Temporin−SHa (**e**,**f**). AMPs at 32, 64, 128, 256, 512 and 1024 µg/mL and experimental control (0). Points: data averages. Error bars: minimum and maximum values. The non-intersection of the error bars denotes a statistical difference according 95% confidence interval (*p* < 0.05) (*n =* 12/group).

**Figure 9 microorganisms-12-00099-f009:**
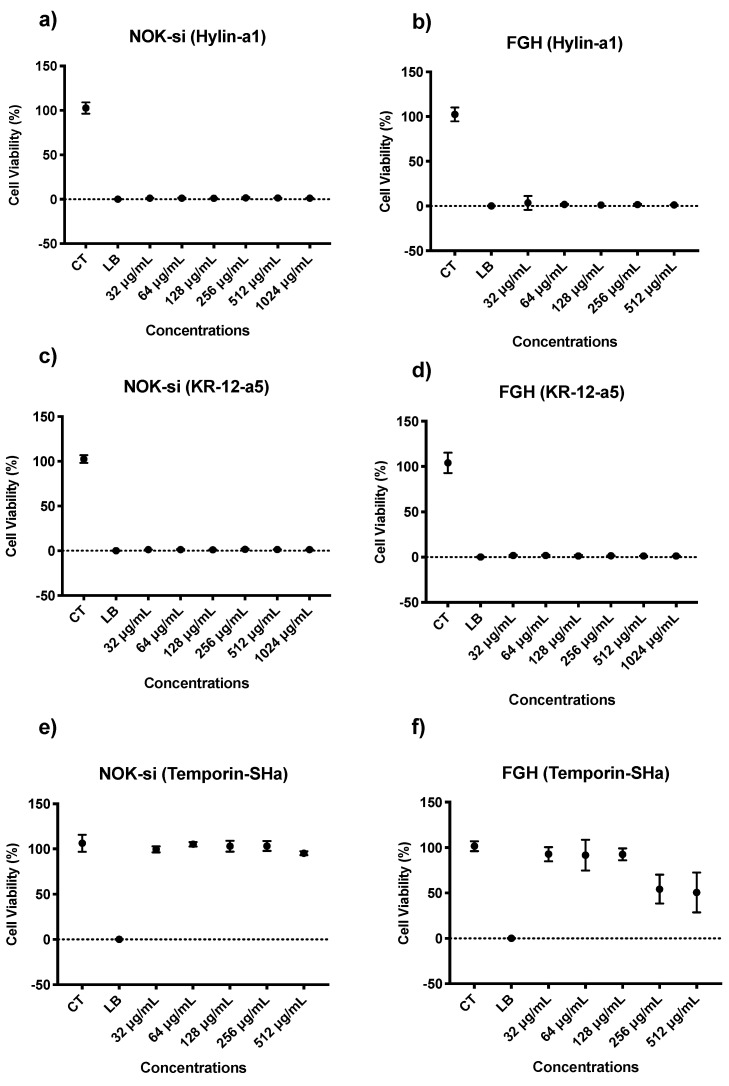
Mean and 95% confidence interval of FGH and NOK−si cellular viability percentage (%) by alamarBlue^TM^ assay after treatment with Hylin−a1 (**a**,**b**), KR−12-a5 (**c**,**d**), and Temporin−SHa (**e**,**f**) at 32, 64, 128, 256, 512, and 1024 µg/mL. CT: live cell control; LB: lysis buffer (dead cell control). Points: data means. Error bars: minimum and maximum values. The non-intersection of error bars denotes the statistical difference according to the 95% confidence interval (*n =* 12/group; *p* < 0.05).

**Figure 10 microorganisms-12-00099-f010:**
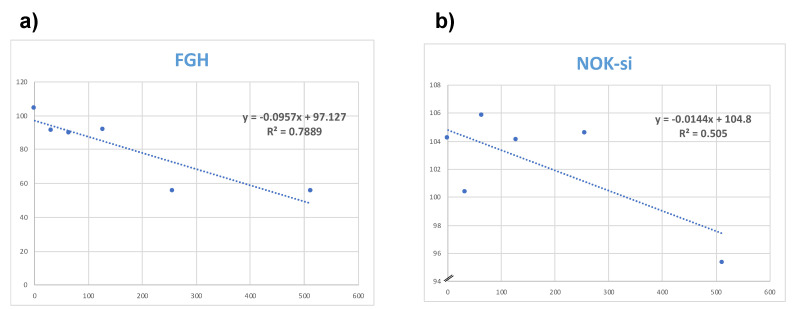
Representative linear regression graph for calculating CC 50%. Y = percentage of living cells; X = concentration of (**a**) Gingival Fibroblasts (FGH) and (**b**) Oral Keratinocytes.

**Table 1 microorganisms-12-00099-t001:** Classification of the cytotoxicity of treatments in relation to control by ISO 10993-5:2009 guidelines.

Classifications	% of Viability
non-cytotoxic	<25%
slightly cytotoxic	25–50%
moderately cytotoxic	50–75%
severely cytotoxic	>75%

## Data Availability

The authors confirm that the data supporting the findings of this study are available within the article and its Appendix A.

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
