# Peer review of "Antibiofilm Activity and Biocompatibility of Temporin-SHa: A Promising Antimicrobial Peptide for Control of Fluconazole-Resistant *Candida albicans"

_microorganisms, 2024, doi:10.3390/microorganisms12010099_

Round 1
Reviewer 1 Report
Comments and Suggestions for Authors
Dear Authors,
The study presented in the present manuscript is focused on the investigation of the effect of the antimicrobial peptides Hlyin-a1, KR-12-a5 and Temporin-SHa in Candida albicans, as well as their biocaompatibility.
There are some issues that I detected while reading your paper:
- in the investigation of the minimum inhibitory/fungicide concentrations, there is no antifungal used for comparation. This is mandatory, if you want to discuss the potential effect against fungal strains, especially on fluconazole-resistant strains;
- you should write NaH2PO4 and Na2HPO4;
- you should write the names of pathogens in Italic;
- in line 141, you wanted to write "µl"?
- In Table 1, there is a different letter font and size;
- the Conclusion part of the manuscript has to be detailed.
The references are well chosen, in agreement with the theme presented.
Comments on the Quality of English Language
- you should write NaH2PO4 and Na2HPO4;
- you should write the names of pathogens in Italic;
- in line 141, you wanted to write "µl"?
- In Table 1, there is a different letter font and size;
Author Response
Dear Reviewer #1,
We thank the reviewers for their valuable time and thoughtfulness in reviewing our manuscript. Enclosed is our revised version entitled " Antibiofilm activity and biocompatibility of Temporin-SHa: A promising antimicrobial peptide to control of fluconazole-resistant Candida albicans". Again, we appreciate the contributions the reviewers' have made. Their input helped us improve our manuscript, and we hope that it is suitable for publication after the revision.
As suggested, we submitted the manuscript for revision (attached), and the changes made in the current submitted manuscript (attached) are addressed in blue color.
Sincerely,
Ana Claudia Pavarina
Walter L. Siqueira
*Corresponding Authors

Reviewer 2 Report
Comments and Suggestions for Authors
1. Introduction needs to be more structured. problem of AMR should be emphasized.
2. Emphasis should be given on Candida biofilm and its role in drug resistance.
3. In the method section; 2.4- bacterial adhesion has been demonstrated at 1.5 h. Authors need to clarify this.
4. Quality of figure 4 is poor. Especially X axis is not visible
5. Since most of the experiments on biofilm have been conducted using values greater than MICs. there are chances of development of resistance as biofilm inhibition certainly because of the reduction of biomass. Authors need to provide a suitable explanation to this.
6. What about the stability of synthesized AMPs
7. results on biofilm needs to be discussed with previous findings.
8. Authors need to provide the possible mechanism of action
9. English needs thorough revision. there are glaring grammatical errors.
Comments on the Quality of English Language
English needs thorough revision
Author Response
Dear Reviewer #2,
We thank the reviewers for their valuable time and thoughtfulness in reviewing our manuscript. Enclosed is our revised version entitled " Antibiofilm activity and biocompatibility of Temporin-SHa: A promising antimicrobial peptide to control of fluconazole-resistant Candida albicans". Again, we appreciate the contributions the reviewers' have made. Their input helped us improve our manuscript, and we hope that it is suitable for publication after the revision.
As suggested, we submitted the manuscript for revision (attached), and the changes made in the current submitted manuscript (attached) are addressed in blue color.
Sincerely,
Ana Claudia Pavarina
Walter L. Siqueira
*Corresponding Authors

Reviewer 3 Report
Comments and Suggestions for Authors
Due to the antimicrobial resistance nature of Candida albicans. Alternative antimicrobials are needed to overcome this challenge. In the present study, the authors have evaluated the antimicrobial activity of antimicrobial peptides (AMPs) Hlyin- 13 a1, KR-12-a5, and Temporin-Sha against sensitive, resistant (CaS and CaR) and biofilm of Candida albicans. Moreover, the cytotoxic effect was evaluated. The study result showed that The AMPs Hylin-a1 and Temporin-SHa have antifungal activity against CaS and CaR; however, only Temporin-SHa has an anti-biofilm effect in both CaS and CaR and is bi- 467 ocompatible on NOK-si and FGH oral cells in monolayer. This study is fascinating, relevant, and well-written. However, some minor issues need to be corrected before considering this manuscript for publication.
Specific Comments and Suggestions for Authors
- Line: 185 Please put the fungal name in italics
- Please cite all methods you used in this study
- Line 255: there was an inconsistency between the result shown in the text and the data shown in the figure. Figure 4- f was mentioned in text, and then showed as F 4 d ? please check
- The abbreviation (hrs or h) must be consistent throughout the whole manuscript.
- Line 308: Figure 6-f instead of Figure 6-d. please check
- Line 325: a concentration of XXXX ????? what concentration
- Line 391: in a previous study of C. albicans……. Please write citations to this study
- in a previous study of C. albicans
- Did the author use positive control when performing antimicrobial testing?
Author Response
Dear Reviewer #3,
We thank the reviewers for their valuable time and thoughtfulness in reviewing our manuscript. Enclosed is our revised version entitled " Antibiofilm activity and biocompatibility of Temporin-SHa: A promising antimicrobial peptide to control of fluconazole-resistant Candida albicans". Again, we appreciate the contributions the reviewers' have made. Their input helped us improve our manuscript, and we hope that it is suitable for publication after the revision.
As suggested, we submitted the manuscript for revision (attached), and the changes made in the current submitted manuscript (attached) are addressed in blue color.
Sincerely,
Ana Claudia Pavarina
Walter L. Siqueira
*Corresponding Authors

Round 2
Reviewer 1 Report
Comments and Suggestions for Authors
Dear Authors,
Thank you for considering my comments and suggestions on your manuscript and for making the changes required.
Author Response
Thank you for the contributions. We appreciate it.